# Resistant Starch Contents of Starch Isolated from Black Longan Seeds

**DOI:** 10.3390/molecules26113405

**Published:** 2021-06-04

**Authors:** Nisit Kittipongpatana, Pairote Wiriyacharee, Rewat Phongphisutthinant, Supakit Chaipoot, Chalermkwan Somjai, Ornanong S. Kittipongpatana

**Affiliations:** 1Department of Pharmaceutical Sciences, Faculty of Pharmacy, Chiang Mai University, Chiang Mai 50200, Thailand; nisit.k@cmu.ac.th; 2Research Center for Agricultural Innovation, Chiang Mai University, Chiang Mai 50200, Thailand; 3Division of Product Development Technology, Faculty of Agro-Industry, Chiang Mai University, Chiang Mai 50100, Thailand; pairote.w@cmu.ac.th (P.W.); chalermkwansomjai@hotmail.com (C.S.); 4Science and Technology Research Institute of Chiang Mai University, Chiang Mai 50200, Thailand; rewat.p@cmu.ac.th (R.P.); supakit.ch@cmu.ac.th (S.C.); 5Center of Excellent in Microbial Diversity and Sustainable Utilization, Faculty of Science, Chiang Mai University, Chiang Mai 50200, Thailand

**Keywords:** resistant starch, black longan, *Dimocarpus longan*, incubation, physicochemical prop-erty, in vitro digestibility

## Abstract

A large quantity of longan fruits (*Dimocarpus longan* Lour.) produced annually are processed into many products, one of which is black longan, from which the dried, dark-brown meat has been used medicinally in traditional medicine, while the starch-containing seeds are discarded. In this study, starch samples (BLGSs) were isolated from seeds of black longan fruits prepared using varied conditions. The in vitro digestibility was determined in comparison with those extracted from fresh (FLGS) and dried (DLGS) seeds. Scanning electron microscopy (SEM), X-ray diffraction (XRD), differential scanning calorimetry (DSC), and Fourier transform infrared (FTIR) spectroscopy were employed to evaluate the starch properties. The results showed that the yields of FLGS, DLGS, and BLGSs were 20%, 23%, and 16–22% *w*/*w*, respectively. SEM images showed starch granules of mixed shapes, with sizes up to 15 µm in all samples. XRD patterns confirmed an A-type crystallinity for FLGS and DLGS, with strong refraction peaks at 2θ = 15°, 17°, 18°, and 23°, while BLGSs also showed detectable peaks at 2θ = 10° and 21°, which suggested V-type structures. Thermal properties corroborated the changes by showing increases in peak gelatinization temperature (T_p_) and enthalpy energy (ΔH) in BLGSs. The paste viscosity of BLGSs (5% *w*/*w*) decreased by 20–58% from that of FLGS. The FTIR peak ratio at 1045/1022 and 1022/995 cm^−1^ also indicated an increase in ordered structure in BLGSs compared to FLGS. The significant increase in the amounts of slowly digestible starch (SDS) and resistant starch (RS) in BLGSs compared to FLGS, especially at a prolonged incubation time of 20 (4.2×) and 30 days (4.1×), was proposed to be due to the heat-induced formation of starch inclusion with other components inside the seed during the black longan production process. Thus, black longan seed could be a new source of starch, with increased RS content, for potential use in the food and related industries.

## 1. Introduction

Longan (*Dimocarpus longan* Lour.) is a popular tropical fruit grown in many parts of Asia. In addition to fresh consumption, longan fruits were processed into many products, ranging from canned longan in syrup to oven-dried longan, all of which subsequently yielded seeds as the by-product. Although longan seeds were widely researched and reported to contain phenolic compounds with many biological activities [1,2], only a small portion was utilized for such aspect, while most of the seeds were discarded as waste or used as fertilizer. Longan seeds contain starch, among other natural components, and because of the large amount of longan agriculturally produced each year, the seed has been viewed as a potentially new commercial source of starch. Hu et al. [3] reported that longan seeds contained up to 49.5% *w*/*w* starch, with high amylose contents (26.6–30.1%), while a study by Guo et al. [4] reported 59.4% *w*/*w* starch, with a 25.1% amylose content. In the production process of dried longan, the whole fruits were placed in a drying chamber of an oven, with controlled temperature and hot air circulation. While the moisture was gradually taken out from the flesh (aril), the seed inside the fruit was subjected to heat via the air and the moisture in the flesh. This created an environment of the heat-moisture treatment (HMT), which is known to alter the thermal and digestibility properties via changes in starch granular arrangement [5]. On the other hand, it is also possible that the seed coat prevented the moisture from getting through and the starch inside the seed was affected only by the dry hot air, causing a different response from that of HMT. In addition to the regular drying of longan fruits, which yielded the “golden meat” dried longan, further incubation at certain humidity and temperature for some amount of time would change the appearance of the longan meat as a result of the Maillard reaction [6]. The conditions employed in the production of this so-called black longan (BLG), of which the darkish brown to black meat has been used medicinally in the practice of Traditional Chinese Medicine (TCM) [2], could potentially affect the properties of starch, including physicochemical, digestibility, and functional properties. Thus, instead of being discarded, the seeds of BLG could be collected, extracted for the starch, and evaluated for the change in digestibility profile, including the contents of rapidly digestible starch (RDS), slowly digestible starch (SDS), and resistant starch (RS). Piecyk et al. [7] suggested that digestion of starch could be altered by many factors, including the presence of intact tissues surrounding starch granules, a high amount of viscous soluble fiber fraction, the presence of compounds that inhibit digestibility, the amylose content, and the interactions between amylose chains. In this study, starch samples were extracted from seeds of BLG prepared by various treatments. The physicochemical properties and in vitro digestibility of BLG seed starch samples were subsequently determined. The results were discussed in comparison with starches extracted from seeds of fresh fruits (FLGS) and seeds of oven-dried fruits (DLGS).

## 2. Results

### 2.1. Extraction and Isolation of Starches from Longan Seeds

Starch yields for FLGS and DLGS were 20.3% and 23.1% DW of seeds respectively, while the yields for BLGS starches ranged between 16% and 22%.

### 2.2. SEM Analysis

SEM images of FLGS, DLGS, and BLGS (Figure 1a–c) showed that granules of longan seed starch were mixtures of oval, polygonal, irregular, and spherical shapes of varied sizes up to 15 µm. No significant difference was observed among the three types of longan starches. SEM of seed marc revealed what appeared to be starch granules clustering together inside.

### 2.3. XRD

X-ray diffraction patterns of FLGS, DLGS, and four representative BLGSs (BLGS-60-5, BLGS-60-10, BLGS-60-20, BLGS-60-30) are shown in Figure 2. All longan starch samples showed strong diffraction peaks at Bragg angle 2θ of 15°, 17°, 18°, and 23°. BLGS samples also exhibited two additional visible peaks at 2θ, of 10° and 20°, which were not detected in XRDs of FLGS and DLGS.

### 2.4. Thermal Properties

The peak gelatinization temperature (Tp), the thermal transition range (To–Tc), and the gelatinization enthalpy (ΔH) of FLGS registered respectively at 76.9 °C, 71.7–85 °C, and 14.9 J/g (Table 1). BLGS samples showed comparable or higher T_p_ values compared to that of FLGS. The thermal transition ranges were broader in samples incubated for 5 and 10 days at all three temperatures, while narrower ranges were recorded in samples incubated for 20 and 30 days. ΔH values of BLGS samples were higher than that of FLGS, with the exception of the samples incubated at each temperature for 30 days in which the ΔH values were significantly lower.

### 2.5. Viscosity

The rheological profiles of FLGS, DLGS, and four representative BLGSs are shown in Figure 3. All samples exhibited pseudoplastic flow with thixotropy. At 5% *w*/*w*, FLGS paste yielded the highest viscosity (0.56 ± 0.01 Pa.s), while the DLGS viscosity was 7% less. BLGS samples showed 20–58% decreases in the viscosity, with BLGS-60-20 being the least viscous paste (0.24 ± 0.02 Pa.s).

### 2.6. Attenuated Total Reflectance Fourier-Transform Infrared Spectroscopy (ATR-FTIR)

The deconvoluted ATR-FTIR spectra of FLGS, DLGS, and four representative BLGSs are shown in Figure 4. The 1045/1022 ratio of DLGS and BLGS samples was significantly higher than that of FLGS (Table 1). All BLGS samples exhibited a higher 1045/1022 ratio than that of DLGS, but with no specific trend regarding the temperature and incubation time. For the 1022/995 ratio, FLGS showed the highest value, which was significantly higher than DLGS. A further decrease in the ratio was observed in BLGS samples with increasing incubation time up to 20 days. The 30-day incubated samples showed an increase in the 1022/995 ratio at all three temperatures.

### 2.7. In Vitro Digestibility and Resistant Starch Content

Starch-isolated fresh longan seeds (FLGS) contained mainly RDS (65%), 15% SDS, and relatively low RS (7.6%) (Table 2). The starch extracted from seeds of oven-dried longan fruits (DLGS) exhibited a significantly lower amount of RDS (45%), and higher amounts of SDS (21%) and RS (19%). The starches isolated from seeds of black longan fruits (BLDGs), which were incubated under different temperatures and time periods, yielded varied amounts of RDS, SDS, and RS in the ranges of 26–35%, 22–27%, and 26–31%, respectively. At each temperature, the amount of RDS decreased, while SDS and RS increased in samples with longer incubation times. The results from temperatures of 60 and 70 °C were not significantly different, but both were superior to the temperature of 80 °C in terms of an increase in the RS content. Analysis of RS standard yielded a value of 55% RS (labeled amount at 52.5%).

Starch-isolated fresh longan seeds (FLGS) contained mainly RDS (65%), 15% SDS, and relatively low RS (7.6%) (Table 2). The starch extracted from seeds of oven-dried longan fruits (DLGS) exhibited a significantly lower amount of RDS (45%), and higher amounts of SDS (21%) and RS (19%). The starches isolated from seeds of black longan fruits (BLDGs), which were incubated under different temperatures and time periods, yielded varied amounts of RDS, SDS, and RS in the ranges of 26–35%, 22–27%, and 26–31%, respectively. At each temperature, the amount of RDS decreased, while SDS and RS increased in samples with longer incubation times. The results from temperatures of 60 and 70 °C were not significantly different, but both were superior to the temperature of 80 °C in terms of an increase in the RS content. Analysis of RS standard yielded a value of 55% RS (labeled amount at 52.5%).

## 3. Discussion

The starch yields (16–23%), calculated from the actual starch obtained in each sample, were significantly lower than those reported by previous studies, i.e., 44.9–49.5% [3] and 59% [4]. A significant amount of starch remained entrapped in the seed marc even after repeated extraction (Figure 1d) and could not be cleanly extracted out without compromising the quality due to contamination with the non-starch components. This portion of starch was disregarded, which contributed to the lower-than-actual starch yield across all samples. Overall, the extracted starches were of good quality and were used in the next experiments. The SEM results were in agreement with previous reports [3,8]. No significant difference in the granule morphology was observed between FLGS and other heat-treated/incubated starches. XRD patterns of longan seed starch were consistent with an A-type crystallinity pattern, as reported previously [3]. DLGS exhibited similar peak intensity to that of FLGS, indicating that the process used in the drying of whole longan fruits did not significantly affect the amorphous/crystalline nature of starch granules. This finding is different from the effect of heat-moisture treatment (HMT), in which the XRD pattern of HMT starch often showed changes in peak intensity as a result of the modification in the granular structure [9]. Thus, the drying of longan fruits to yield oven-dried longan was not classified as HMT. In contrast, diffraction peaks with stronger intensities were observed in BLGS samples, especially those isolated from seeds of black longan fruits incubated at longer incubation times, which suggested that elevated temperature (60–80 °C) and time of exposure played a role in altering the inner structure of starch. Two additional peaks visibly detected in BLGSs at Bragg angle 2θ of 10° and 20° could be related to the V-type crystal structure of starch inclusion complexes which were originated from the heat-activated interaction between amylose and other components inside the seeds. Longan seeds contained a number of long-chain fatty acids (e.g., palmitic acid, oleic acid) and polyphenolic compounds, including tannins (e.g., gallic acid, ellagic acid, corilagin, chebulagic acid, geraniin) [10,11]. Polyphenolic compounds are known to non-covalently interact with starch through hydrophobic interaction, forming inclusion complexes which can be confirmed by the presence of a V-type pattern in XRD. The inclusion complex showed resistance to enzyme digestion [12]. DSC thermal parameters were in line with the values reported previously [3,4]. DLGS exhibited a slightly broader thermal transition range and a higher ΔH, suggesting that the heat-drying process has brought about a degree of molecular order. The increase in gelatinization temperature could be due to the formation of inclusion complexes between starch and other components in the seed [13,14]. The higher values represent the formation of an ordered structure in starch granules, thus confirming the results observed in XRD. Although the formation of starch inclusion complex usually takes place at high temperatures, specifically during the gelatinization of starch [14], a study by Ahmadi-Abhari et al. [15] suggested that the formation of amylose inclusion complex could also take place and complete at temperatures between 50 and 60 °C when allowed prolonged incubation time, without the starch being gelatinized. The decrease in paste viscosity observed in BLGS samples was consistent with that reported for the starch–lipid complex [16]. This could partially be due to the increase in pasting temperature of BLGSs, which reduced the melting and leeching of starch granules [17]. The ratio of absorbance at 1045/1022 cm^−1^ obtained from the deconvoluted ATR-FTIR was known as an effective means to characterize the degree of order in starch granules [18], while the ratio at 1022/995 cm^−1^ was suggested to correlate with the formation of a double helix in starch molecules [19]. The higher ratio of peaks 1045/1022 observed in DLGS and BLGS samples compared to that of FLGS indicated that heat-drying and further incubation increased the ordered crystalline structure in starch granules. The study by Hu et al. [3] showed that raw longan starches of three varieties underwent digestion at 15–20% after 20 min, and at 60% after 2 h of exposure to digestion enzymes. Although the amount of resistant starch was not reported, the digestogram showed that the digestion approached 100% after 7 h. Amylose content and granule crystallinity were cited as major factors that impeded enzyme penetration into the granules, thus slowing down the digestion. The exposure of seeds to heat applied for drying while still inside the fruits could alter the physicochemical properties of the starch inside the seed “pocket”, similar to the effects of dry heat on starch granule [20], although to a smaller magnitude. The heat could also simply cause the shrinkage of the granules, therefore limiting the access of enzymes into the granules and retarding the digestion.

## 4. Materials and Methods

### 4.1. Materials

Longan fruits, grade B, were obtained from a local supplier (Honeyqueen Co., Ltd., Lamphun, Thailand). The seeds of fresh fruits were acquired by manual peeling, washed, and air-dried before subjecting to starch extraction. The Resistant Starch Assay Kit (AOAC Method 2002.02) was a product of Megazyme (Wicklow, Ireland). All organic solvents and chemicals used were reagent grade, unless indicated otherwise.

### 4.2. Methods

#### 4.2.1. Oven-Drying of Longan Fruits

Whole longan fruits were dried in a hot-air oven using a proprietary drying condition of Honeyqueen Co., Ltd. (Lamphun, Thailand). For quality control purpose, the pulp of the resulting dried whole longan fruits was analyzed to contain 14–17% moisture content and 0.5–0.6 water activity (aw) according to the Thai Agricultural Standard (TAS 10-2006) [21]. Seeds were taken from a portion of the dried longan fruits for starch extraction.

#### 4.2.2. Incubation of Dry Longan to Produce Black Longan

Subsequently, twelve (12) sets of 500 g of oven-dried, whole longan fruits (control sample) were incubated at 60, 70, and 80 °C, in desiccators previously equilibrated at approximately 75% relative humidity using saturated NaCl solution, for 5, 10, 20, and 30 days. At the end, the seeds of incubated “black” longan samples were separated from the pulp/meat, washed, and collected for starch extraction.

#### 4.2.3. Extraction and Isolation of Starch from Longan Seeds

All seed samples were subjected to starch extraction using a method described previously for Jackfruit seed starch [22], with the additional NaCl solution/toluene cleanup wash as reported by Hu et al. [3]. Starch samples obtained from fresh seeds and oven-dried seeds were labeled FLGS (fresh longan seed starch) and DLGS (dry longan seed starch), respectively. Starch samples of incubated “black” longan seeds were tagged BLGS (black longan seed starch), followed by the temperature and days of incubation.

#### 4.2.4. Scanning Electron Microscopic (SEM) Analysis

SEM experiments to analyze the granule surface, shape, and size were conducted using a JEOL instrument, model JSM-5410LV (JEOL, Peabody, MA, USA), equipped with a large field detector. The acceleration voltage was 15 kV under low-vacuum mode (0.7–0.8 torr). The sample was placed on a copper stub covered with adhesive tape and coated with gold under vacuum. The images were taken at 1000X magnification.

#### 4.2.5. X-Ray Diffraction (XRD)

XRD patterns were recorded in the reflection mode on a Siemens D-500 X-ray diffractometer (Siemens, Munich, Germany). Diffractograms were registered at a Bragg angle (2θ) range of 5–40° at a scan rate of 2.5°/min and step size of 0.02°.

#### 4.2.6. Attenuated Total Reflectance Fourier-Transform Infrared Spectroscopy (ATR-FTIR)

FTIR spectra were recorded on a Nicolet Nexus 470 FTIR equipped with a DTGS detector (Thermo Fisher Scientific Inc., Waltham, MA, USA) using an attenuated total reflectance (ATR) mode. For each spectrum, 64 scans were recorded at a resolution of 4 cm^−1^. Spectra were baseline-corrected using Omnic ver.6.2. The region at 1200–900 cm^−1^ was deconvoluted and the absorbance values at 1045, 1022, and 995 cm^−1^ were determined. The peak ratios of 1045/1022 and 1022/995 were calculated for each sample.

#### 4.2.7. Thermal Properties

Thermal properties were assessed using a Perkin Elmer DSC-7 (Waltham, MA, USA) differential scanning calorimeter. The analysis was carried out at a temperature range of 30–120 °C, at 10 °C/min, on a 1:3 (*w*/*w*) starch–water mixture sample. An empty pan was used as a reference. The temperatures of the characteristic transitions, including onset (To), peak (Tp), and conclusion (Tc) temperatures, were recorded. Enthalpy change of gelatinization (ΔH) was calculated and expressed as J/g of dry starch.

#### 4.2.8. Viscosity

Apparent viscosity of longan starch paste was determined using a Brookfield R/S-CPS rheometer (Bob-and-Cup format, from Toronto, ON, Canada). The samples were prepared by dispersing 2.5 g of starch sample in 50 mL distilled water, mixed thoroughly, and stir-heated at 90 °C for 15 min. The measuring system was CC48 DIN. The mode used was CSR (controlled shear rate). The measured parameters consisted of three steps: (1) an increase of the shear rate from 0 to 100 s^−1^ in 1 min, (2) held at 100 s^−1^ for 1 min, and (3) a decrease of the shear rate from 100 to 0 s^−1^ in 1 min. All measurements were performed in triplicate, at a controlled temperature of 25 ± 1 °C. The data were analyzed with Brookfield Rheo 2000 software. The apparent viscosity for all samples in this study was measured at a shear rate of 100 s^−1^ and was expressed in Pa s.

#### 4.2.9. In Vitro Digestibility of Longan Starches

The digestibility of starches obtained from fresh, dry, and black longan seeds was analyzed using a Megazyme Resistant Starch Assay Kit (AOAC Method 2002.02) according to a procedure previously described [23]. In brief, a set of three screw-capped test tubes containing 100 mg of sample and 4 mL of solution of pancreatic α-amylase (PPA) (10 mg/mL, pH 6) and amyloglucosidase (AAG) (3 U/mL) were prepared for each sample. The tubes were then incubated in a shaking water bath set at 37 °C. A reaction tube was collected for each sample after 20 min, 2 h, and 16 h of incubation, for determination of rapidly digestible starch (RDS), slowly digestible starch (SDS), and resistant starch (RS), respectively. The reaction was stopped by adding 4 mL of ethanol and centrifuged at 4000× g for 10 min to separate the digested (supernatant) part from the non-digested (residue) part. The supernatant was diluted with 100 mM sodium acetate buffer. An aliquot of the solution was incubated with amyloglucosidase (10 μL, 300 U/mL) at 50 °C for 20 min. The residue was dissolved in 2 M KOH (2 mL) in an ice bath, added with 1.2 M sodium acetate buffer (8 mL), and hydrolyzed to glucose with amyloglucosidase (0.1 mL, 3300 U/mL) at 50 °C for 30 min. The glucose oxidase/peroxidase (GOPOD) reagent was added to the aliquot portion of each part and the mixture was incubated at 50 °C for 20 min. Absorbance of the mixture was then measured at 510 nm. Resistant starch and non-resistant (digested) starch were calculated as glucose × 0.9. The total starch was calculated as the sum of resistant and digested starch. RDS, SDS, and RS fractions were calculated as percentage of total starch at each time period. A standard starch of known RS percentage and a commercial RS starch, Hi-Maize 260, were also determined under the same condition for comparison purposes.

### 4.3. Statistical Analysis

All tests were performed at least in triplicate. The statistical significance tests were performed using analysis of variance (ANOVA) at the 95% confidence level (*p* < 0.05). Significant differences among mean values were determined by Duncan’s multiple range test.

## 5. Conclusions

The enormous amount of longan seeds from the fresh fruits and the dried longan industry, together with previous reports on starch isolation, suggested that this biowaste could potentially be a new source of commercial starch. The starches extracted from black longan seeds showed similar granule size and structure but exhibited different in vitro digestibility profiles compared to those extracted from seeds of fresh or oven-dried fruits. The amounts of resistant starch found in BLGS samples were 4× higher than that of FLGS. Additional peaks in XRD, transition parameters in DSC, and decreased paste viscosity of BLGS suggested that the increases in resistant starch content were possibly due to the starch inclusion complexes, which were formed through interaction with other components in the seeds during the temperature/time-controlled incubation. The optimum temperatures and time for incubation of dried longan fruits to obtain black longan seeds with high RS content were 60–70 °C for 20–30 days. The applied heat altered starch properties in a way similar to that caused by dry heat treatment. Black longan seed starch could be a new source of resistant starch for applications in the food, pharmaceutical, and related industries.

## Figures and Tables

**Figure 1 molecules-26-03405-f001:**
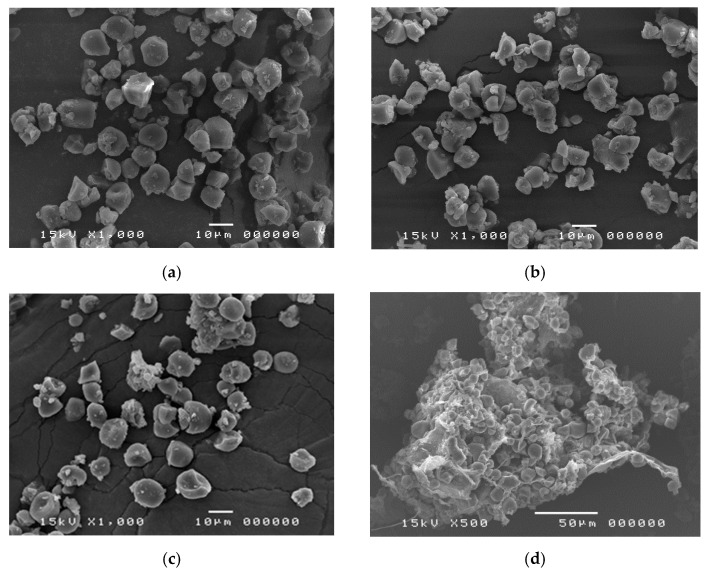
Representative SEM images of longan starches extracted from (**a**) seeds of fresh fruits (FLGS), (**b**) oven-dried fruits (DLGS), (**c**) “black longans” fruits, and (**d**) longan seed marc after extraction, showing that many starch granules remained intact.

**Figure 2 molecules-26-03405-f002:**
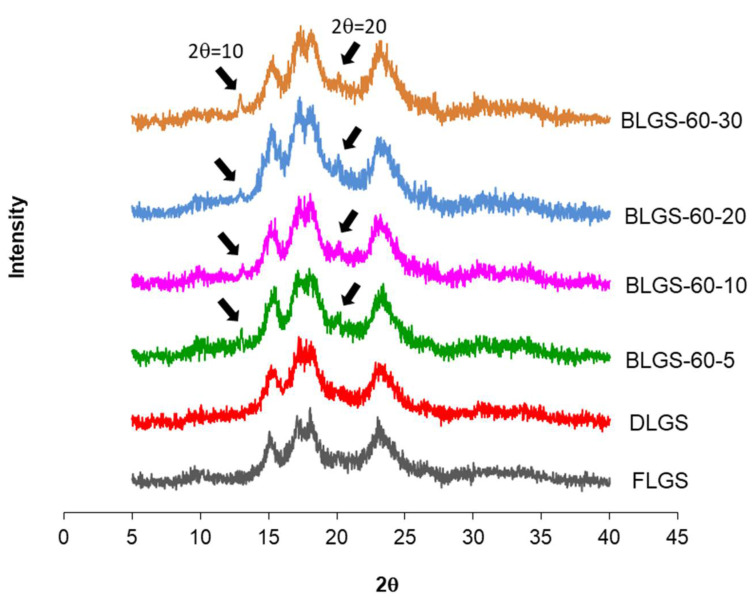
Representative XRD images of longan starches extracted from seeds of fresh fruits (FLGS), oven-dried fruits (DLGS), and “black longan” fruits incubated at 60 °C for 5, 10, 20, and 30 days (BLGS-60-5, BLGS-60-10, BLGS-60-20, BLGS-60-30).

**Figure 3 molecules-26-03405-f003:**
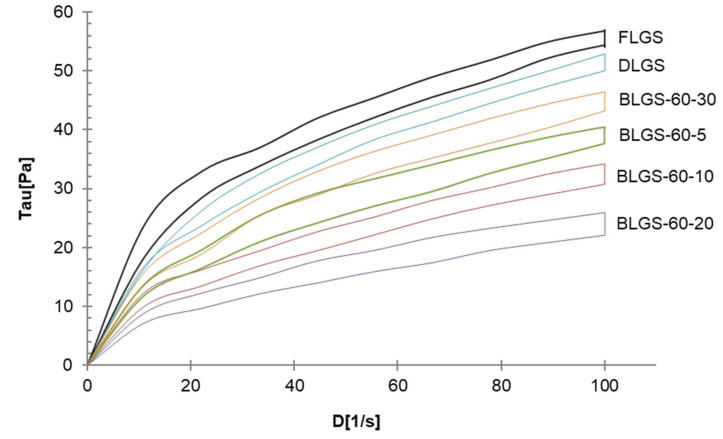
Representative rheological profiles of longan seed starches of varied materials.

**Figure 4 molecules-26-03405-f004:**
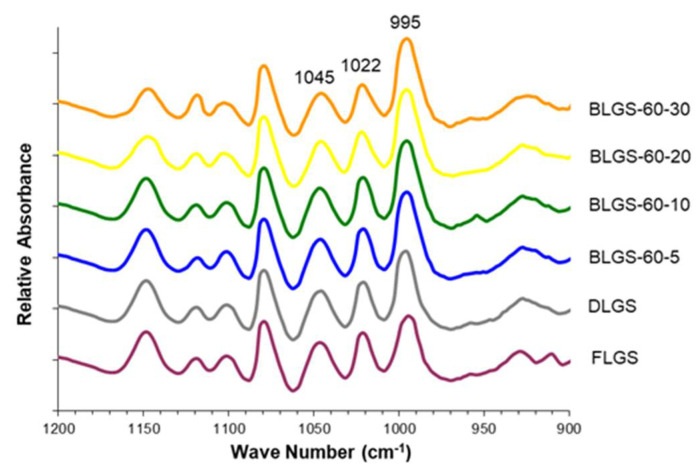
Representative ATR-FTIR spectra of longan seed starches of varied materials.

**Table 1 molecules-26-03405-t001:** Thermal properties and IR peak ratios of longan seed starch of different preparation conditions.

Sample	Temperature (°C)			ΔH (J/g)	IR Peak Ratio	
	T_o_	T_p_	T_c_		1045/1022	1022/995
FLGS	71.7 ± 0.7 ^a^	76.9 ± 0.3 ^b,c^	85.0 ± 1.2 ^a^	14.9 ± 0.6 ^b,c^	0.553	0.610
DLGS	70.6 ± 0.4 ^c^	76.9 ± 0.2 ^b,c^	84.9 ± 0.6 ^a,b^	15.4 ± 1.6 ^a^	0.588	0.459
BLGS-60-5	70.9 ± 0.3 ^b,c^	77.3 ± 0.2 ^a,b^	85.4 ± 1.1 ^a^	15.1 ± 0.8 ^a,b^	0.607	0.438
BLGS-60-10	71.1 ± 0.6 ^b^	77.7 ± 0.3 ^a^	84.6 ± 0.8 ^a,b^	15.8 ± 1.4 ^a^	0.635	0.430
BLGS-60-20	71.6 ± 0.2 ^a^	76.3 ± 0.8 ^c^	84.7 ± 0.8 ^a,b^	14.3 ± 1.7 ^c,d^	0.676	0.297
BLGS-60-30	71.8 ± 0.6 ^a^	77.6 ± 0.4 ^a,b^	83.2 ± 1.4 ^b,c^	13.5 ± 1.4 ^d^	0.629	0.389
BLGS-70-5	70.2 ± 0.5 ^e^	77.6 ± 0.5 ^a,b^	85.2 ± 1.1 ^a^	15.8 ± 0.6 ^a^	0.651	0.447
BLGS-70-10	71.4 ± 0.5 ^a,b^	76.7 ± 0.4 ^c^	83.5 ± 0.7 ^b,c^	14.7 ± 0.6 ^b,c^	0.617	0.425
BLGS-70-20	71.8 ± 0.9 ^a^	77.2 ± 0.4 ^b^	85.4 ± 1 ^a^	15.1 ± 1.1 ^a,b^	0.633	0.407
BLGS-70-30	72.5 ± 0.6 ^a^	77.4 ± 0.6 ^a,b^	84.1 ± 0.5 ^a,b^	13.2 ± 1.2 ^d^	0.598	0.452
BLGS-80-5	70.6 ± 0.3 ^c^	75.3 ± 0.6 ^d^	81.7 ± 0.8 ^d^	15.5 ± 1.1 ^a^	0.604	0.481
BLGS-80-10	71.1 ± 0.9 ^b^	76.9 ± 0.3 ^c^	84.7 ± 0.8 ^a,b^	15.6 ± 0.9 ^a^	0.595	0.462
BLGS-80-20	71.9 ± 0.4 ^a^	77.8 ± 0.3 ^a^	84.7 ± 0.8 ^a,b^	14.7 ± 0.9 ^b,c^	0.633	0.439
BLGS-80-30	72.8 ± 1 ^a^	78.0 ± 0.6 ^a^	84.4 ± 1.1 ^a,b^	14.1 ± 0.4 ^c,d^	0.647	0.489

Data are mean ± SD. Means within the same column with different letters indicate significant difference (*p* ≤ 0.05) by Duncan’s multiple range test.

**Table 2 molecules-26-03405-t002:** Rapidly digestible starch (RDS), slowly digestible starch (SDS), and resistant starch (RS) contents of fresh longan seed starch (FLGS), dried longan seed starch (DLGS), and dried, incubated (“black”) longan seed starches (BLGS) prepared under varied conditions.

Sample *	Drying/Incubation Condition		In Vitro Digestibility (%) **			Total Starch (%) **
	Temp (°C)	Days	RDS	SDS	RS	
FLGS	-	-	65.64 ± 3.90 ^a^	15.32 ± 1.92 ^d^	7.56 ± 1.02 ^f^	88.52 ± 1.05 ^a^
DLGS	14–17% MC, 0.5–0.6 aw		45.34 ± 2.86 ^b^	21.21 ± 1.76 ^c^	19.50 ± 4.46 ^e^	86.05 ± 6.60 ^b^
BLGS-60-5	60	5	35.11 ± 1.64 ^c^	22.32 ± 1.73 ^c^	26.54 ± 1.32 ^c^	83.97 ± 2.62 ^b,c^
BLGS-60-10	60	10	33.21 ± 1.77 ^c^	22.87 ± 0.32 ^c^	28.69 ± 1.52 ^a,b^	84.77 ± 3.45 ^b,c^
BLGS-60-20	60	20	28.64 ± 2.91 ^d^	23.32 ± 2.35 ^b,c^	31.24 ± 1.99 ^a^	83.20 ± 2.64 ^b,c^
BLGS-60-30	60	30	27.44 ± 2.21 ^d,e^	25.56 ± 1.02 ^a,b^	30.81 ± 1.61 ^a^	83.81 ± 4.60 ^b,c^
BLGS-70-5	70	5	32.65 ± 1.18 ^c,d^	22.48 ± 1.96 ^c^	27.02 ± 2.29 ^b,c^	82.15 ± 3.81 ^c,d^
BLGS-70-10	70	10	31.15 ± 1.02 ^d^	22.55 ± 0.79 ^c^	28.76 ± 2.08 ^a,b^	82.46 ± 2.23 ^c^
BLGS-70-20	70	20	27.76 ± 3.78 ^d,e^	23.89 ± 1.73 ^b,c^	31.69 ± 2.12 ^a^	83.34 ± 2.73 ^b,c^
BLGS-70-30	70	30	26.69 ± 2.07 ^e^	24.76 ± 1.49 ^b^	31.21 ± 1.20 ^a^	82.66 ± 1.14 ^c^
BLGS-80-5	80	5	33.45 ± 1.94 ^c^	22.63 ± 1.11 ^c^	24.27 ± 1.88 ^d^	80.35 ± 1.19 ^e^
BLGS-80-10	80	10	30.54 ± 2.49 ^d^	24.69 ± 1.84 ^b^	26.44 ± 1.08 ^c^	81.67 ± 0.59 ^d,e^
BLGS-80-20	80	20	27.53 ± 1.72 ^d,e^	26.43 ± 2.60 ^a^	27.18 ± 1.53 ^b,c^	81.14 ± 1.19 ^d,e^
BLGS-80-30	80	30	26.59 ± 0.92 ^e^	27.72 ± 1.35 ^a^	27.87 ± 1.25 ^b,c^	82.18 ± 1.74 ^c,d^
RS Standard	-	-	7.17 ± 0.34	34.85 ± 1.15	55.45 ± 0.79	97.47 ± 0.35

* FLGS—starch from fresh seeds; DLGS—starch from oven-dried seeds; BLGS—starch from incubated seeds. ** All values were reported on a dry basis of starch. Data are mean ± SD. Means within the same column with different letters indicate significant difference (*p* ≤ 0.05) by Duncan’s multiple range test.

## Data Availability

Not applicable.

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
