# Peer review of "Resistant Starch Contents of Starch Isolated from Black Longan Seeds"

_molecules, 2021, doi:10.3390/molecules26113405_

Round 1

Reviewer 1 Report

The manuscript falls into the scope of this journal. In this manuscript, Longyan seed starch was used as the research object to analyze the structure and digestive characteristics of the dried black longyan seed starch and normal longyan seed starch.

   However, the manuscript still has some issues to solve.

  1. The line width is too thick, please adjust it in Figure 2.
  2. What causes a large amount of starch to remain in the dregs of seeds.
  3. line 237, line 244, line 250 and line 256, Please adjust the format and indent the first line.
  4. It is hoped that the author can further isolate and detect the starch inclusion complex, and verify the type and structural characteristics of the inclusion complex.

Author Response

1. The line width is too thick, please adjust it in Figure 2.

Response: The line width was adjusted to a thinner size.

2. What causes a large amount of starch to remain in the dregs of seeds.

Response: The method used in the extraction of starch from seeds was mechanical and chemically mild. This is to ascertain the quality of the starch granules for the study and to prevent contamination from non-starch substances. So we sacrificed the yield for the better starch quality. The statement “A significant amount of starch remained entrapped in the seed marc even after repeated extraction (Figure 1d) and could not be extracted out cleanly without compromising the quality due to contamination with the non-starch components” was to provide the reason for the lower starch yield in our study compared to previous reports.

3. line 237, line 244, line 250 and line 256, Please adjust the format and indent the first line.

Response: The format and indent have been adjusted for those four lines.

4. It is hoped that the author can further isolate and detect the starch inclusion complex, and verify the type and structural characteristics of the inclusion complex.

Response: It is the intention of the authors to report in this paper the existence of starch inclusion complex in black longan starch based on the physical properties and the significant increase in resistant starch content. The isolation and structural characterization of the starch inclusion complex will be attempted in the continuing project and will be reported in due time.

Reviewer 2 Report

The manuscript presents a study on starch from residual seeds, which is relevant. Although the analyzes performed are classic in characterizing starch, the authors did not refer to the viscosity characteristics of the studied starch, which would have been important to compare with other traditional sources. Therefore, it is suggested to the authors that they include some consideration about the viscosity of the studied starch.

Author Response

It is suggested to the authors that they include some consideration about the viscosity of the studied starch.

Response: We have included the viscosity study on the longan starch paste (5%w/v) in the revised version of the manuscript (section 2.5 in Results and 4.2.8 in Methods). Consequential changes (additions) have been made in the abstract, discussion, conclusions and references.